# Dynamic Seed Emission, Dispersion, and Deposition from Horseweed (*Conyza canadensis* (L.) *Cronquist*)

**DOI:** 10.3390/plants11091102

**Published:** 2022-04-19

**Authors:** Jun Liu, Qidi Zhao, Haiyan Huang, Rongjian Ye, Charles Neal Stewart, Junming Wang

**Affiliations:** 1School of Information Science and Engineering, Chongqing Jiaotong University, Chongqing 400074, China; niuniu77@cqjtu.edu.cn (J.L.); 622210070045@mails.cqjtu.edu.cn (Q.Z.); 2Illinois State Water Survey, Prairie Research Institute, University of Illinois at Urbana-Champaign, Champaign, IL 61820, USA; huanghaiyan.usa@gmail.com; 3Department of Plant Sciences, The University of Tennessee, Knoxville, TN 37996, USA; yerongjian@sinochem.com (R.Y.); nealstewart@utk.edu (C.N.S.)

**Keywords:** seed release rate, concentration, deposition, seed release pattern, seed dispersion pattern

## Abstract

The wide dispersion of glyphosate-resistant (GR) horseweed (*Conyza canadensis* (L.) *Cronquist**: synonym Erigeron canadensis* L.) biotypes has been reported in agricultural fields in many states. GR traits may be transferred through seeds or pollen from fields with existing GR horseweed prevalence to surrounding fields. Understanding seed production and movement is essential when characterizing and predicting the spread of GR horseweed, yet a literature review indicates that there are no experimental data on dynamic (hourly) seed production and horizontal dispersion and deposition from horseweed. To obtain the dynamic data, two field experiments were performed, one in Illinois and one in Tennessee, USA in 2013 and 2014, respectively. Seed concentration and deposition along with atmospheric conditions were measured with samplers in the Illinois (184 m × 46 m, natural plants, density = 9.5 plants/m^2^) and Tennessee (6 m × 6 m, cultivated plants, density = 4 plants/m^2^) experimental fields and their surrounding areas along the downwind direction up to 1 km horizontally and 100 m vertically in the Illinois field and up to 32 m horizontally and 5 m vertically in the Tennessee field. The dynamic seed source strengths (emission rates) measured during two entire seed-shedding seasons were reported, ranging from 0 to 0.41 grains/plant/s for Illinois and ranging from 0 to 0.56 grains/plant/s for Tennessee. The average total seed production was an estimated 122,178 grains/plant for the duration of the Illinois experiment and 94,146 grains/plant for Tennessee. Seeds trapped by Rotorod samplers attached beneath two balloons in the Illinois field experiment were observed at heights of 80 to 100 m, indicating the possibility of long-distance transport. Normalized (by source data) seed deposition with distance followed a negative power exponential function. Seed emission and transport were affected mainly by wind speed. This study is the first to investigate dynamic horseweed seed emission, dispersion, and deposition for an entire seed-shedding season. The results will aid in the management of GR horseweed. The potential for regional effects of horseweed invasion may require all farmers to control horseweed in their individual fields.

## 1. Introduction

Horseweed (*Conyza canadensis* (L.) *Cronquist:*
*synonym Erigeron canadensis* L.) is an annual weed belonging to the aster family (Asteraceae). Glyphosate-resistant (GR) evolved horseweed has been documented in 24 states across the United States and in 10 countries worldwide, and it is an invasive species in some countries (Glyphosate is a nonselective systemic herbicide). In recent years, GR horseweed has become a major agricultural problem across much of North America [1].

Horseweed seedlings first develop basal rosettes and start bolting in mid-April. They bloom from the end of July or the beginning of August [2], and individual flowers are small white ray flowers and yellow disk flowers [3] (Figure 1b). Before the capitula are fully opened, horseweed pollen is released. Horseweed is self-fertile and there are no significant insect herbivores (no significant insect pollination) [2,4,5]. Horseweed plants can reach a height of 2 m, and a single plant can produce nearly 200,000 seeds [6,7] (Figure 1a). The seeds are small achenes (1.6–6.4 mm long) with a pappus of tan to white bristles [8] (Figure 1c). Horseweed seeds are lightweight, weighing approximately 0.7 mg per seed, or 1,400,000 seeds per kg [9]. The seeds have a gravitational-settlement velocity of 0.323 m/s [10,11].

Strategies used to control the spread of horseweed, especially the GR biotypes, across agricultural fields or natural lands require an increased understanding of the seed dispersion process. Many factors can influence seed dispersal distance, including seed source strength (emission rate), meteorological conditions, and topography. Several experimental studies have been conducted on vertical seed dispersal in the air [12,13], but none have focused on dynamic seed release rates and the distance of seed dispersion and deposition. In particular, little information is available on the correlation between seed release rates, horizontal dispersion, and deposition and atmospheric conditions (wind speed, direction, wind variability, and atmospheric stability).

The objectives of the present study were to: (1) measure dynamic (on the order of an hour) horseweed seed emissions under constantly changing atmospheric conditions; (2) measure hourly horseweed seed dispersion and deposition within the near and far scope of the release spot and in the vertical (up to 100 m) and horizontal (up to 1000 m) directions; and (3) analyze the correlation between horseweed seed emission, dispersion, deposition, and atmospheric parameters.

## 2. Results

### 2.1. Seed Production and Emission Dynamics

In the Illinois field containing a large-area of natural horseweed plants, 15,239 seeds were collected in 132 sampling periods inside and outside of the source field during the 44-day experiment. The Tennessee field experiment lasted 41 days, and 3690 seeds were collected in 83 sampling periods. The amounts of the collected seeds were different between the experimental sites and years, which might be caused by the different plant densities. However, horseweed seeds have a similar seasonal release pattern. The release of horseweed seeds lasted approximately 45 days (Figure 2). For the two experiments, seed production started around mid- or late August, and few seeds were released until 3 September. The seed release rate gradually increased and reached its peak around 11 September. The maximum average source strength was 0.41 grains/plant/s for Illinois and 0.56 grains/plant/s for Tennessee. Following that date, the release rate decreased gradually, and after 12 days, on 24 September, the release rate decreased to a very low point. The low release rate continued into mid-October.

The source strength in Illinois ranged from 0 to 9234 grains/plant/day (08:00–20:00) and in Tennessee ranged from 0 to 12,976 grains/plant/day. After the peak rate around 11 September, the release rate declined sharply to a low value in Tennessee; in contrast, a relatively high source strength occurred on some days in Illinois.

Because, during the daytime, the atmosphere is unstable and dry and is favorable for seed release and transport, seed release happens during the daytime, and there is no or little release during the nighttime. The diurnal seed release rate during the main release period (from 3 September to 24 September) of the two fields followed a relatively similar pattern (Figure 3). Seed release mainly started after 09:00, then gradually increased and reached its peak near 14:00; subsequently, it decreased sharply and remained low during the late afternoon. In Illinois, the daily release pattern curve was relatively smooth in the morning (i.e., from 09:00 to 13:00), and the maximum hourly source strength was approximately 0.41 grains/plant/s, which occurred near 14:00 on 11 September. In Tennessee, the daily release pattern curve from 09:00 to 12:00 was sharp, the source strength was relatively low compared to Illinois, and the maximum release rate was 0.56 grains/plant/s at 14:00 on 10 September. All the times are local time.

Following Wang and Yang’s method [14], seed source strength was composed of integrated horizontal flux density and vertical flux density. As shown in Figure 4, the contribution of integrated horizontal flux density to the source strength ranged from 0 to 36% for Illinois and 0 to 52% for Tennessee.

### 2.2. Concentration with Height and Distance

Measurements of seed dispersal patterns are important for horseweed transportation research. The seed concentration decreased sharply with height and distance.

In Tennessee, the concentration variation with low height (0–5 m) followed a negative logarithmic function (Figure 5). The concentration was the highest at the lowest height and decreased rapidly with height. In the Illinois field, the concentration variation with high height (0–100 m) followed a negative power function (Figure 6). The rapid variation occurred from the ground level up to 5 m. From 10 m to 100 m, the concentration decreased slowly, and the variation was small.

The horizontal dispersion of seed was analyzed along the downwind direction. In the Tennessee field, normalized concentration (normalized by the concentration at 0 m from the edge of the field) with distance followed a negative natural exponential function (Figure 7). A sharp reduction occurred at 0–2 m, and the average concentration was 0.06 grains/$m^{3}$, 0.03 grains/$m^{3}$ at 1 m and 2 m, respectively.

In Illinois, with the balloon measurement, seeds were found at heights of 80 to 100 m at 0–10% of the source concentration, which was approximately 0 to 0.08 grains/m^3^, revealing that seeds can be dispersed to a high altitude and potentially to a long distance. At a 20 to 40 m downwind distance of the Illinois source field and at lower heights (<10 m), the seed concentration ranged from 2 to 90% (0.03 to 0.94 grains/m^3^) of the source concentration (Figure 8). At farther distances of 40–70 m at low heights (<10 m), 2 to 20% (0.03 to 0.47 grains/m^3^) of seed concentration remained. At the 70–150 m distance in the downwind direction, many concentrations in the air at all heights were approximately 0 to 5% of the concentration in the Illinois source field (approximately 0 to 0.08 grains/m^3^).

### 2.3. Deposition with Distance

Seed deposition with distance followed a negative power exponential curve in the two field experiments (Figure 9). In the Tennessee field, the deposition rapidly decreased to 4% (0.003 grains/$m^{2}$/s) at a 4 m distance from the source field, then gradually decreased with distance. In the Illinois field, the deposition decreased to 2% (0.0008 grains/$m^{2}$/s) at 35 m from the source field, then gradually decreased with distance. At 320 m, the average deposition was 1.8% (0.0025 grains/$m^{2}$/s). At 1000 m, seeds were not detected using the small sampling tile.

### 2.4. Influence of Meteorological Factors

Pearson correlation coefficients (r) between seed and meteorological parameters were calculated. Multiple tests were conducted on the same data, so an adjustment to the α-value was made to avoid type I errors. The α-value was divided by the number of tests (k), α’ = α/k, to accomplish a Bonferroni correction. In this study, α was assumed to be 0.05 and k was 10; therefore, the new significance level threshold α’ was 0.005.

#### 2.4.1. Source Production

In the Illinois field, source strength was moderately and positively correlated with horizontal mean wind speed $\bar{u_{1}}$ (|r| > 0.45 and <0.7, *p* < 0.005), weakly correlated with solar radiation (|r| ≤ 0.45), and not significantly (*p* > 0.005) correlated with other atmospheric parameters. This may explain the fact that the field center concentrations, deposition, and IHF at the source were significantly related to wind speed (*p* < 0.005).

A regression model (using stepwise regression) was obtained to simulate source strength per plant (*SS*) based on meteorological parameters. The regression equation is:(1)$$ss=0.471{\bar{U}}_{1}+0.265SR,R^{2}=0.26\;\left(p<0.005\right)$$

This equation shows that the variation of source production is affected mainly by wind speed and solar radiation. Solar radiation may have a biological effect on seed release. Stronger solar radiation may have dried plant tissues, resulting in lighter seeds that release more easily.

Figure 10 also shows the correlation between the source strength, wind speed, and solar radiation. Seed source strength usually varied with the solar radiation. When solar radiation began to increase, horseweed plants started releasing more seeds. Furthermore, the seed release peaks usually occurred during the high wind speed time. As a result, the highest peak of the seed release rate during the day was during high solar radiation and wind speed, such as on 11 September, 10:30–13:30 in Illinois and 10 September, 15:20–16:44 in Tennessee.

#### 2.4.2. Seed Transport

The horizontal deposition ratio (deposition at different downwind distances to source strength) with the downwind distance was correlated with wind and variations of wind speed in both experiments. The deposition research in the Tennessee field was in the range of 0 to 32 m from the edge of the release field. At the source field edge (i.e., 0 m), the horizontal deposition ratio was significantly correlated with source strength and wind speed (|r| > 0.50 and <0.7, *p* < 0.005). The Illinois field experiment was correlated with wind speed, variations of wind speed $\sigma_{u1}$ (distance > 40 and distance < 80 m) (|r| > 0.45 and <0.7, *p* < 0.005), and u* (distance > 80 and distance < 160 m) (|r| > 0.45 and <0.7, *p* < 0.005).

In the Illinois field experiment, correlation analysis suggested that vertical transport at a high altitude was strongly correlated with air temperature at heights of 60–100 m (|r| > 0.7, *p* < 0.005). Thus, the primary atmospheric parameter affecting vertical transport may have been vertical turbulence (air temperature is correlated with turbulence). At heights of 0–50 m in the Illinois field experiment, seed concentration was positively related to wind speed $\bar{U_{1}}\left(3.3\right)$ (|r| ≤ 0.45, *p* < 0.005) and u (|r| ≤ 0.45, *p* < 0.005). This implies that stronger wind may have transported more seeds farther.

## 3. Discussion

### 3.1. Source Strength

Two factors are commonly used as determinants of the seed dispersal process: seed weight and production abundance. As mentioned in the introduction, horseweed seed is lightweight, so it can travel easily with the wind. Horseweed has also been documented as a plant with a relatively large seed production. For example, Loux et al. (2014) [8] indicated that a single horseweed plant can produce up to 200,000 seeds. Davis et al. (2009) [15] showed that seed production is also a function of plant height, and the magnitude of seed production from different biotypes varies from 10,000 to 100,000 seeds per plant. In the study carried out by Steckel (2014) [16], seed production ranged from 50,000 to 250,000 seeds/plant. In this study, the estimated average total number of seeds produced by each plant was 122,178 for Illinois and 94,146 for Tennessee. Compared with the literature, the observation of seed production in this study was reasonable. In Illinois, the major seed release days numbered approximately 34 days, occurring from 6 September to 8 October (Figure 2a). On other days, the seed release ranged from 0 to 0.06 grains/plant/s, which was approximately 0–15% of the peak day release. In Tennessee, the main release period was slightly shorter than in Illinois, approximately 22 days, from 3 September to 24 September (Figure 2b). On other days, seed release ranged from 0 to 0.05 grains/plant/s, which was approximately 0–8% of the peak day release.

The effects of diurnal fluctuations on seed production have been investigated extensively for different species [17,18,19]. Generally, diurnal fluctuations are caused by pollinator activity, solar radiation cycles, high-low ambient temperature differences, and atmospheric stability. The diurnal seed release pattern shown in Figure 3 was reasonable. The peak release was around 14:00 when the solar radiation was high, relative humidity was low, and wind speed and turbulence were strong. These atmospheric conditions made seed release and transport easier. The correlation analysis and the regression equation for source strength also showed that strong horizontal wind and solar radiation mainly affected seed releases.

### 3.2. High Altitude and Long-Distance Transport

Various studies have estimated the mechanics of seed dispersal by wind and elucidated the relative importance of physical and biological factors that affect seed dispersal. It has long been recognized that release height is an essential factor in the seed dispersal process [20,21,22]. Previous studies indicate that to some extent the heights of particles can determine their dispersal range. Shields et al. (2006) [13] reported that horseweed seed concentrations at a height of 80 m were 0.0001–0.001 grains/$m^{3}$. In this study, the concentration range of 0–0.03 grains/$m^{3}$ at the altitude of 60–100 m in the Illinois field experiment was greater than that in their experiments. The differences may be caused by source strength, source field size, atmospheric conditions, and sampling methods.

As shown in Figure 9, most of the seeds fell within 200 m of the Illinois source field. At a long distance from the Illinois source field, such as ~480 m, seeds were still detected, even though the deposition rate was relatively low (0.01 grains/$m^{2}$/s on a peak release day, or 36 grains/$m^{2}$/hour). This can pose a serious weed spread problem during a seed-shedding season. At 1000 m, no seeds were found on deposition tiles in the Illinois source field.

Although the dispersal distance of seed is influenced by many aspects of plant biology, including the spreading of invasive species, metapopulation dynamics, and diversity and dynamics in plant communities [23], there are few data sets that characterize the exact dispersal distances because of the limited detection at long distances and at low deposition conditions. The presence of seeds at a high altitude (80–100 m) in the Illinois source field experiment illustrates that horseweed plant seeds were capable of being transported a long distance. If seeds were lifted to the height of 80 m, and the settlement velocity was 0.323 m/s [10,11], then seeds would require 248 s to reach the ground. At noon, the seeds could easily travel 1.24 km with a light wind of 5 m/s. Simulated horseweed seed dispersion using the Hysplit model [24] showed that some seeds should be transported beyond 1000 m. However, in the sampling periods at the Illinois horizontal 1000 m location, the samplers caught no seeds. This may have resulted from the small sampler size, the specific sampling location, and/or the amount of sampling time. The dispersal distance deserves more explicit explorations using models and other experimental methods.

### 3.3. Influence of Meteorological Factors

#### 3.3.1. Source Strength

Information on seed release rates is important for understanding and predicting seed dispersal. Several authors have noted that seed release rates vary with respect to seed ripening and environmental conditions such as wind speed, turbulence, and air humidity [25,26]. The favorable meteorological conditions that can promote seed release include low humidity, high temperature, unstable atmosphere, strong wind, and little precipitation [27,28]. The effects of meteorological factors on the release of seed may differ depending on the local climatic features and topography, as well as the type of plant. As expected, positive correlations were observed between source strength and wind speed and source strength and solar radiation.

This result is quite reasonable because, as suggested by many other studies, high wind speeds and turbulence can promote the abscission of seeds from plants [26,27,28]. Solar radiation tends to positively correlate with both concentration and deposition, thus favoring source strength. At the same time, it was also observed that the correlation between relative humidity and concentrations and deposition at the source was negative. It has long been recognized that high relative humidity can physically prevent abscission by hindering the opening of the involucres or promoting the closing of the drag-producing fibres, resulting in fewer seeds released [29].

#### 3.3.2. Seed Transport

As expected, the horizontal transport correlated with horizontal wind speed in both experiments. Similar to our study, the importance of wind in determining the dispersal distance was noted by Raynor et al. (1972) [30] and Jarosz et al. (2005) [31]. The seed travel distance increased with a higher wind speed.

Because seed transport is influenced by different atmospheric parameters (especially horizontal wind speeds), seed settling speed, canopy structure, and topography, it is impossible to conduct all experiments under all parameter conditions. Modeling work is needed to include these parameters and simulate all possible conditions. Data obtained in the study can be used to calibrate and validate dispersion models. This study provided a complete data set of meteorological parameters and seed release, dispersion, and deposition rates. After models have been validated, analysis of the sensitivity of meteorological parameters on seed dispersion and deposition can be conducted. Then, data on how far seed can be transported under different meteorological conditions can be obtained to complement the measured data.

The high altitude of the seed flight implies long-distance transport. Long-distance transport information is important to policy-making concerning the spread of GR horseweed. For example, the potential for regional effects of horseweed invasion may require all farmers to control horseweed in their individual fields.

## 4. Materials and Methods

### 4.1. Experimental Field Setup

Two experiments were performed in the USA at different locations and in different years: Illinois in 2013 and Tennessee in 2014. Besides measuring the dynamic yield of seeds, the field experiments were needed to evaluate the dispersion effects under different situations; therefore, the sizes and shapes of the two plots were designed differently. The measurements were conducted within a 184 × 46 m plot with natural horseweed plants in Illinois (Figure 11) and a 6 × 6 m plot with cultivated horseweed plants in Tennessee (Figure 12).

In Illinois, seed measurements were conducted from 23 August to 12 October 2013 on the South Research Farm, University of Illinois at Urbana-Champaign, Champaign, IL, USA (Latitude: 40°04′51.36′′ N; Longitude: 88°14′23.92′′ W; Elevation: 216 m). The experimental source field was full of naturally occurring horseweed plants with a density of 9.5 plants/$m^{2}$. Various grasses and soybean surrounded the source field. The average canopy height of the horseweed was 1 m. The plants were fully mature and flowering by 23 August and started to release seeds on 29 August. The prevailing wind direction was from the southwest to the northeast. Sampling occurred in the downwind prevailing direction.

In Tennessee, horseweed seed emission and dispersion sampling were conducted from 18 August 2014 to 1 October 2014 at the University of Tennessee, Knoxville, Tennessee, USA (Latitude: 35°53′46.57′′ N; Longitude: 83°57′35.99′′ W; Elevation: 250 m, Figure 12). The source plants were GR horseweed cultivated in a circular area with a diameter of 12 m. The density was 4 plants/$m^{2}$. The mean plant height was 1.3 m. The prevailing wind direction during the experimental period was from the northeast to the southwest.

Both field experiments were designed to measure the hourly source strength of horseweed plants. Seed concentration and deposition decrease dramatically with distance and height with a negative power law or logarithm function [32,33]. Therefore, the Illinois field experiment was designed mainly for measuring the seed dispersion over a long distance and high altitude. The release plot was a large area (8464 $m^{2}$) with a high plant density (9.5 plants/$m^{2}$). However, in the Tennessee field, a larger number of samplers were placed close to the source field and ground to measure the concentration and deposition of seeds at a short distance and a lower height. The experimental source plants were cultivated in a 12 m diameter circular area with a low density of 4 plants/$m^{2}$. Table 1 summarizes all measurements made in the two fields.

### 4.2. Seed Concentration

Seed concentration (grains/$m^{3}$) could be used to calculate the release rate (i.e., the source strength) and reflect the dispersion of horseweed seeds. Seed concentration was measured by Rotorod samplers (Model 20, Sampling Technologies, Inc., Minneapolis, MN, USA) inside or outside the fields. The slides were coated with silicon grease and set up on Rotorod samplers (Figure 13b). The collection efficiency of the slide was assumed to be 100% and independent of wind speeds [33].

Seed vertical concentration (grains/$m^{3}$) profiles intended for calculating the seed release rate, were usually measured at the field centers. In Illinois, one-column Rotorod samplers (RC1, Figure 11) worked for this. These Rotorod samplers were mounted at heights of 0.35 m, 1 m (i.e., the height of the canopy), 1.7 m, and 2.8 m above the ground (Figure 11b,d). In Tennessee, a column of five Rotorod samplers (HP1, Figure 12) was mounted on a mast in the center of the circular field. These samplers were at a height of 0.5 m (i.e., height of the flowers), 1.3 m (i.e., height of the canopy), 2.23 m, 3.15 m, and 5 m above the ground (Figure 12b). Meanwhile, wind speeds were measured to estimate the horizontal flux density. In addition, additional recording samplers were set up in the center of the source fields at the canopy height (1 m in Illinois and 1.3 m in Tennessee) to measure the seed release rate uninterruptedly (CR, Figure 11 and Figure 12), which was the reference value of the source strength when the other samplers were switched off to insert fresh slides.

Seed concentration along the downwind direction could reflect the dispersion of the seeds with the wind. In Illinois, one-column Rotorod samplers (RC2, Figure 11) were set up at the northeast edge of the source field. Samplers were at 1, 1.5, and 3 m heights. Another two columns of Rotorod samplers (CB1, CB2, Figure 11) were mounted beneath two balloons to collect the concentration data at greater heights, ranging from 10 to 100 m. The two balloons floated in the downwind direction, one inside and one outside of the field at a maximum distance of 1000 m downwind and away from the edge of the source field. The downwind distance of the balloons and the sampler heights were adjusted depending on whether seeds were detectable. Outside of the field in Tennessee, a series of Rotorod slides were placed 0.5 m high at several distances downwind from the source field (HR9-HR12, HR14, HR16, Figure 12) to observe seed concentration dispersion. Besides these samplers, two masts were set up at 4 and 16 m downwind from the edge of the source plot, with Rotorod samplers at 0.5, 1.3, 3.15, and 5 m heights (HP2, HP3, Figure 12).

Horseweed seeds that adhered to the slides were approximately 1.6–6.4 mm long and were easy to distinguish from other debris and to count. The seed concentration (C, grains/$m^{3}$) was calculated as the total sampled seed number on each slide divided by the sampled air volume calculated as a function of the rotation speed, sampling time, and slide area of the corresponding Rotorod sampler [34].
(2)$$C=\frac{N}{∆V\times \omega \times ∆T}$$

In Equation (2), N was the number of seeds on the slide, ∆V ($m^{3}$) was the air volume sampled by the slide during each revolution, ω  (revolution/s) was the rotating speed of the Rotorod sampler, and ∆T (s) was the duration of each sampling period. When combined with wind speed measurements and concentration, the horizontal flux (grains/$m^{2}$/s) profiles could be obtained, indicating the amount of seed production and release.

### 4.3. Seed Deposition

The vertical flux of seeds (deposition, grains/$m^{2}$/s) in the source fields was measured using greased microscope slides/tiles placed near the ground along the downwind direction. Slide sizes were the same as those used in the Rotorod samplers, 75 × 25 mm. Outside of the source fields, the greased tiles used to gather seeds were bigger than those used inside the field (tile size: 150 × 150 mm). The locations of deposition slides/tiles at the two experimental fields were shown in Figure 11 and Figure 12. The deposition slides and tiles were placed and collected at the same time as the concentration slides. The collection efficiency of deposition slides and tiles was assumed to be 100% [30,33,35]. There were no overloading problems during the sampling periods on the deposition slides and tiles.

The seed deposition rate (D, grains/$m^{2}$/s) was determined by the total number of seeds on the counting areas of each slide and tile, divided by the sampling area and time.

During rainy days, plants did not release seeds, so the experiments were not conducted on those days.

Each concentration and deposition sampling period in Illinois and Tennessee was 1.5–3 h between 08:00 and 20:00 local time. The microscope slides/tiles were replaced with a fresh set each sampling period between 08:00 and 20:00.

### 4.4. Meteorological Parameters

The atmospheric parameters including wind velocity, solar radiation, air temperature, relative humidity, and rainfall were recorded during the whole pollination period in both experiments. The three-dimensional wind velocities and air temperatures were measured by sonic anemometers (CSAT3, Campbell Scientific, Logan, UT, USA). In the Illinois field experiment, a sonic anemometer was placed 3.3 m above the ground, and three-dimensional wind velocities and air temperatures were recorded at 10 Hz using a CR3000 data logger (Campbell Scientific). In the Tennessee field, a sonic anemometer was placed at a height of 2.6 m above the ground in the center of the field.

The other weather data were provided by weather stations. In Illinois, the weather station was approximately 800 m north of the source field. The Water and Atmospheric Resources Monitoring Program at the Illinois State Water Survey, University of Illinois at Urbana-Champaign records solar radiation, air temperature, relative humidity, and rainfall every hour. In Tennessee, a weather station (Davis Vantage Pro2TM Weather Station, Davis Instruments, Inc., Hayward, CA, USA) was located about 10 m north of the source field.

The meteorological parameters that denote the atmospheric conditions were estimated by using measurements of weather stations and sonic anemometers for each sampling period. All raw instantaneous velocity components (u, v, w) from sonic anemometers were averaged over 30 min, and the mean values of wind velocities ($\overset{¯}{u},\overset{¯}{v},\overset{¯}{w}$) were calculated. The raw data were rotated to a new coordinate system, i.e., the v-axis was rotated to $\overset{¯}{v}$ = 0, the w-axis was rotated to $\overset{¯}{w}$ = 0, and the u-axis was rotated with the v- and w-axes, accordingly. After rotation, the atmospheric parameters were recalculated, including mean wind speed ($\bar{u_{1}}$, m/s), wind direction (Θ, degree), friction velocity (u*, m/s), and atmospheric stability (ξ, unitless) (Table 2).

$\bar{u_{1}}$ The friction velocity (u*, m/s) was obtained from the formulation [36]:(3)$$u^{*}=\sqrt[{4}]{{\left(cov\left(u_{1},w_{1}\right)\right)}^{2}+{\left(cov\left(v_{1},w_{1}\right)\right)}^{2}}$$
where u_1_, v_1_, and w_1_ are the wind instantaneous velocity components after the rotation, and $cov\left(u_{1},w_{1}\right)$ and $cov\left(v_{1},w_{1}\right)$ were the covariances between $u_{1}$ (or $v_{1}$) and $w_{1}$. To calculate atmospheric stability (ξ, unitless), the Monin-Obukhov length (L, m) parameter was calculated first, according to the equation [36]:(4)$$L=-\frac{\bar{T}\times u^{*}{}^{3}}{k\times g\times cov\left(T,w_{1}\right)}$$
where $\overset{¯}{T}$ was the mean air temperature, u* (m/s) was the friction velocity, k was the von Kármán constant (k was assumed to be 0.4), g was the gravitational acceleration (9.8 m/$s^{2}$), and $cov\left(T,w_{1}\right)$ was the covariance between the anemometer instantaneous temperature and the rotated vertical velocity. The calculation of the atmospheric stability (ξ, unitless) was performed using equation [36]:(5)ξ=z/L
where z was the height (m), and L was the parameter of Monin-Obukhov length (L, m). Z was set to 3.3 and 2.6 m, respectively, for Illinois and Tennessee.

Based on these parameters, the standard deviations of mean wind speed ($\sigma_{u1}$, m/s) and vertical wind velocity ($\sigma_{w1}$, m/s) were also calculated. The standard deviation of vertical wind velocity was used to represent turbulence strength [36].

### 4.5. Source Strength

Source strength was a significant indicator of the amount of seeds produced per unit area or plant per unit time. Because the settling speed of horseweed seed (0.323 m/s) [10,32] was relatively large, the dynamic source strength of horseweed seed should account for both the horizontal and vertical flux densities. Therefore, the source strength included both the integrated horizontal flux (IHF, grains/$m^{2}$/s) and the downward deposition flux densities (D, grains/$m^{2}/s$) of seed grains [33].

The integrated horizontal flux (IHF) was used to indicate the horizontal flux density of seeds [14,37] and was calculated by integrating the product of seed concentration at height z, C (z) (grains/$m^{3}$), and the wind speed at height z ($\overset{¯}{u}\left(z\right)$, m/s) during each sampling period (Figure 13a). The trapezoidal method was applied in the integration. According to the atmospheric similarity theory, wind speeds at different heights were calculated by inputting u*, L, and height (z) [36].
(6)$$U\left(z\right)=\{\begin{array}{c} pro\left(z_{0},z_{0}\right)e^{2\left(\frac{z}{z_{0}}-1\right)},z\leq z_{0}\\ pro\left(z,z_{0}\right),z>z_{0}\;\;\;\;\;\;\;\;\;\;\;\;\;\;\;\; \end{array}$$
where $z_{0}$ is the height of the canopy, and pro($z_{1}$, $z_{2}$) is defined as
(7)$$pro\left(z_{1},z_{2}\right)=\frac{5u^{*}}{2}\left[\ln\left(\frac{10z_{1}}{z_{2}}\right)-phi\left(z_{1}\right)+phi\left(\frac{z_{2}}{10}\right)\right]$$
where phi (*z*) is defined as:(8)$$phi\left(z\right)=\{\begin{array}{c} 2ln\frac{1+{\left(1-\frac{15z}{L}\right)}^{\frac{1}{4}}}{2}\\ -\frac{5z}{L},L\geq 0 \end{array}+ln\frac{1+{\left(1-\frac{15z}{L}\right)}^{\frac{1}{2}}}{2}-2arctan{\left(1-\frac{15z}{L}\right)}^{\frac{1}{4}}+\frac{\pi }{2}\times L<0\;$$

Consequently, the source strength, $Q_{0}\;$ (grains/$m^{2}$/s), was calculated as the summation of IHF divided by the contribution plant number (product of area and plant density) and the deposition divided by the density.
(9)$$Q_{0}=\frac{\int_{0}^{R}D\left(r\right)\mathrm{dr}+}{R}+\frac{\int_{0}^{Z}C\left(z\right)U\left(z\right)\mathrm{dz}}{R}$$
where D (r) was the deposition flux density (grains/$m^{2}$/s), r was the distance from the field center along the wind direction, and R was the maximum distance between the field edge to the center of the field along the wind direction. For the 2013 Illinois field, R = 63.9 m, and for the 2014 Tennessee field, R = 6 m; C (z) (grains/$m^{3}$) was the concentration of seeds at the center of the field at height z (m), U (z) (m/s) was the wind speed at height z (m), and Z was the maximum height of the concentration samplers.

$Q_{0}\;$ (grains/$m^{2}$/s) can be converted to units (grains/plant/s) by dividing plant density (plants/$m^{2}$).

### 4.6. Correlation

Correlation analyses were conducted to reveal the complicated influences that atmospheric parameters have on horseweed seed emission, dispersion, and deposition. Atmospheric parameters were $\bar{u_{1}}$, u, ξ, mean air temperature ($\overset{¯}{T}$), mean solar radiation (SR), and mean relative humidity (RH), etc. (Table 2). Seed dispersal parameters included seed concentration and deposition in and outside of the field. All data analyses were fulfilled with the commercial software Minitabrelease 13 (Minitab, 2000, State College, PA, USA).

## 5. Conclusions

This is the first study that obtained data on the dynamic source strength (emission rate) of horseweed seed during entire seed-shedding seasons in different years and locations. The average total seed production was an estimated 122,178 grains/plant/whole season for the Illinois fields and 94,146 for the Tennessee fields. A regression equation was obtained to determine source strength based on atmospheric parameters. Horseweed seeds were observed reaching heights of 80 to 100 m, indicating that long-distance transport is possible. Normalized seed concentration with height followed a negative logarithmic function when height ranged from 0 to 5 m and a negative power law function when height ranged from 0 to 100 m. Normalized seed deposition with distance followed a negative power exponential function. Seed emissions and transport were mainly affected by wind speed. The potential for regional effects of horseweed invasion may require all farmers to control horseweed in their individual fields.

## Figures and Tables

**Figure 1 plants-11-01102-f001:**
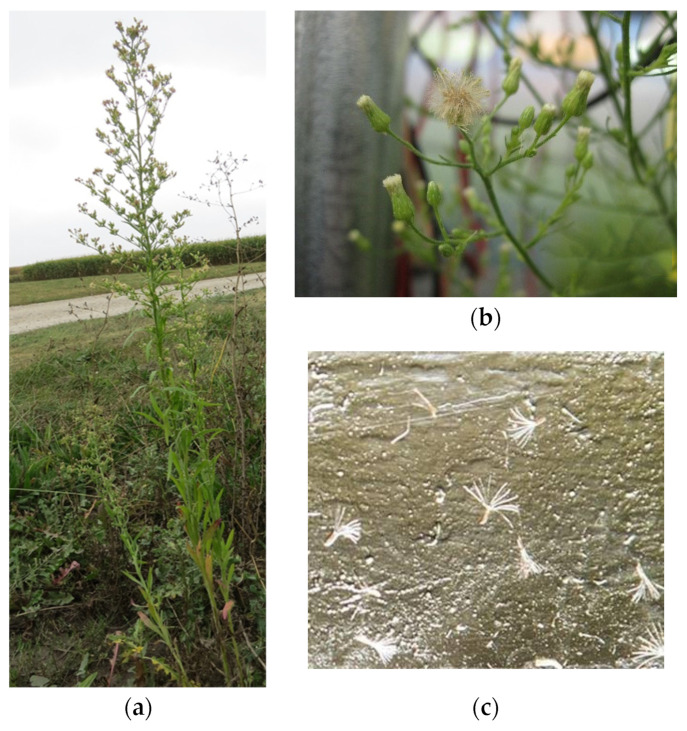
Structure of horseweed plants. (**a**) Mature horseweed plants; (**b**) Flowers of horseweed plant; (**c**) Horseweed seeds in a microscopic field.

**Figure 2 plants-11-01102-f002:**
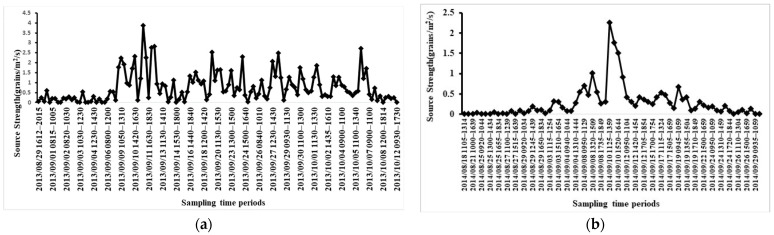
Seasonal seed source strength in the two experiments. (**a**) Hourly average seed source strength measured in the Illinois source field from 29 August 2013 to 12 October 2013; (**b**) hourly average seed source strength measured in the Tennessee source field from 18 August 2014 to 29 September 2014; (**c**) estimated daily released grains per plant (08:00 to 20:00) during the main release period (3 September to 24 September) for Illinois (black solid line) and Tennessee (grey dotted line).

**Figure 3 plants-11-01102-f003:**
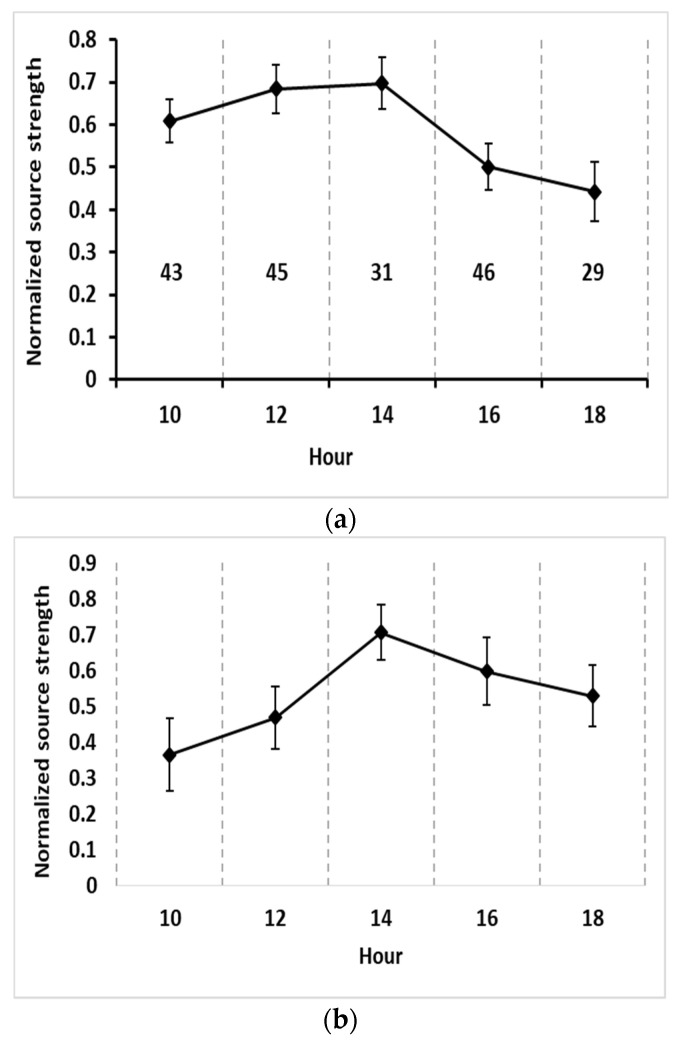
Diurnal variation of normalized seed source strength (normalized by daily maximum; the diamonds are means and bars are standard deviations). Numbers above the x-axis are the number of samples used for the corresponding calculations of mean and standard deviation. (**a**) The normalized seed source strength of the Illinois source field from 3 September 2013 to 24 September 2013; (**b**) The normalized seed source strength of the Tennessee field from 3 September 2014 to 24 September 2014.

**Figure 4 plants-11-01102-f004:**
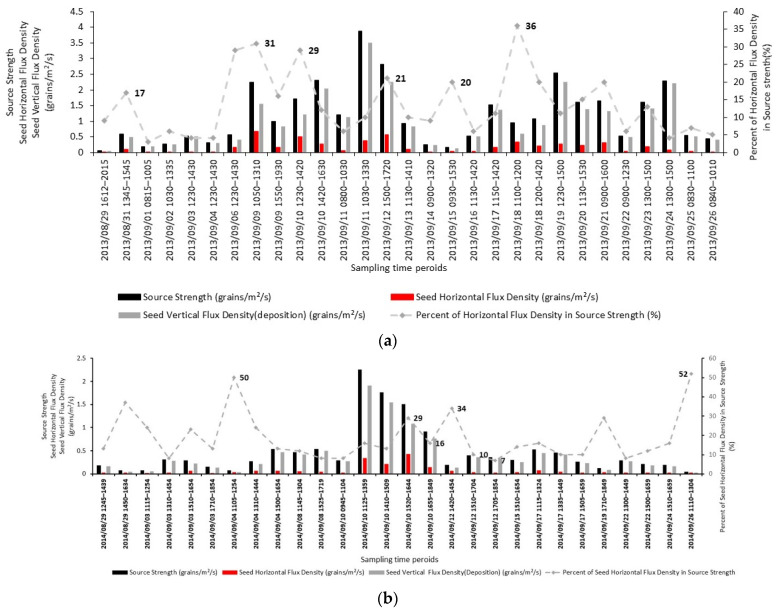
Composition of source strength at the (**a**) Illinois source field; (**b**) Tennessee source field. Source strength (grains/m^2^/s) for every sampling period was composed of the integrated horizontal and vertical flux density. The black bars were the source strength, grey bars were the vertical flux density (first term in Equation (9)), and the red bars were the seed horizontal flux density (second term in Equation (9)). Dotted lines were the percentage of horizontal flux density in seed source strength.

**Figure 5 plants-11-01102-f005:**
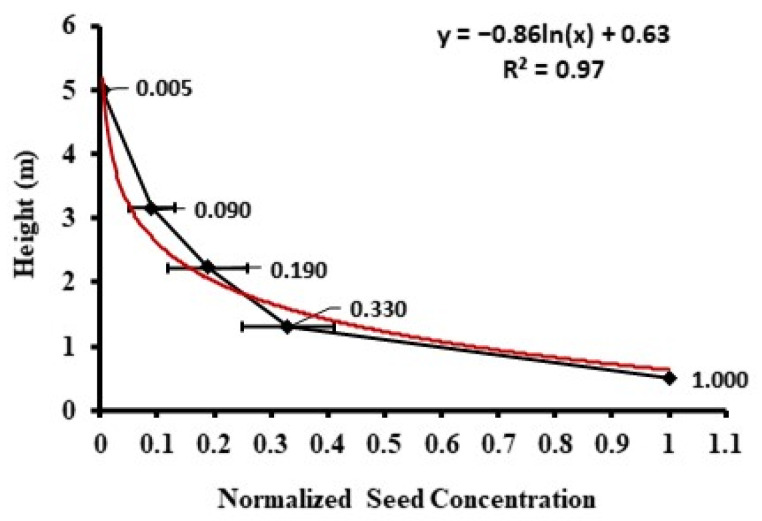
Vertical distribution of normalized seed concentration between the height of 0 and 5 m in the Tennessee field (normalized by the concentration at the center of the field at 0.5 m height). The diamonds are means, and the bars are standard deviations.

**Figure 6 plants-11-01102-f006:**
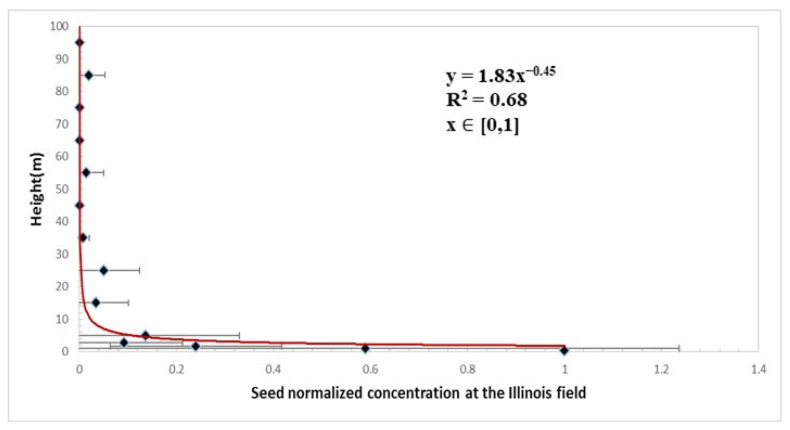
Vertical distribution of normalized seed concentration between the height of 0 and 100 m in the Illinois field (normalized by the concentration at the center of the field at 0.35 m height). The diamonds are means, and the bars are standard deviations.

**Figure 7 plants-11-01102-f007:**
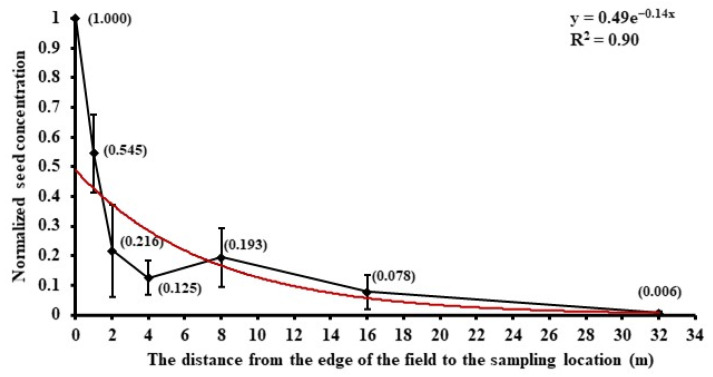
Horizontal dispersion of normalized seed concentration in the Tennessee field (normalized by the concentration at the distance of 0 m from the edge of the field) along the downwind direction, and all the collecting rotorod samplers were set up at the height of 0.5 m. The diamonds are means and bars are standard deviations.

**Figure 8 plants-11-01102-f008:**
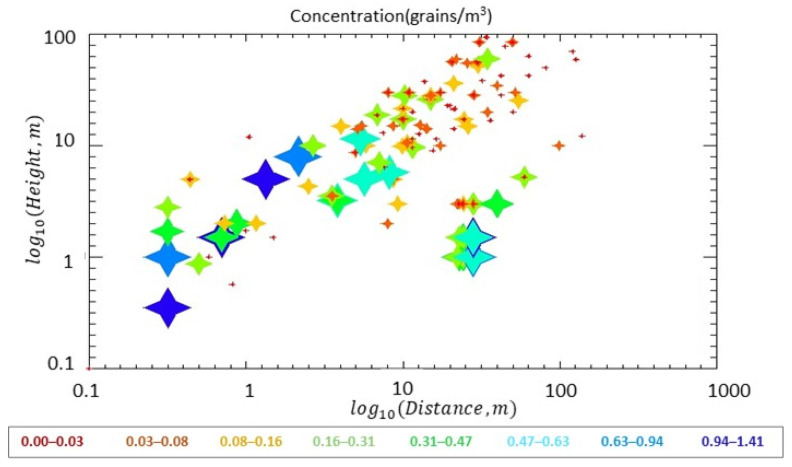
Variation of seed concentration (grains/m^3^) with height and downwind distance in the Illinois field. Points at the downwind distance = 0 m were set as distance = 0.3 m for logarithm scale plotting.

**Figure 9 plants-11-01102-f009:**
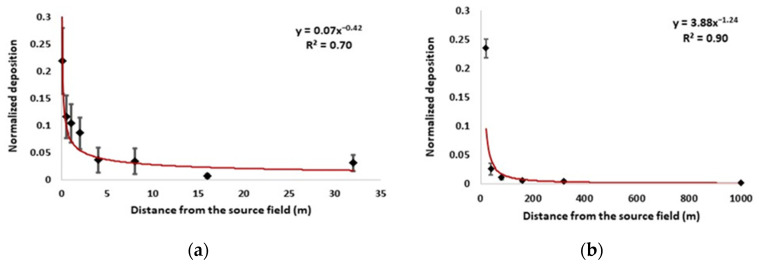
Normalized seed deposition along the downwind direction in the field experiments; x-axis was distance from the source field edge (seed deposition was normalized by the source strength; the diamonds are means and bars are standard deviations; red line is the fitted trend line). (**a**) Normalized seed deposition along the downwind direction in the Tennessee field from 0 to 32 m away from the field edge; (**b**) Normalized seed deposition along the downwind direction in the Illinois field from 20 to 1000 m away from the field edge.

**Figure 10 plants-11-01102-f010:**
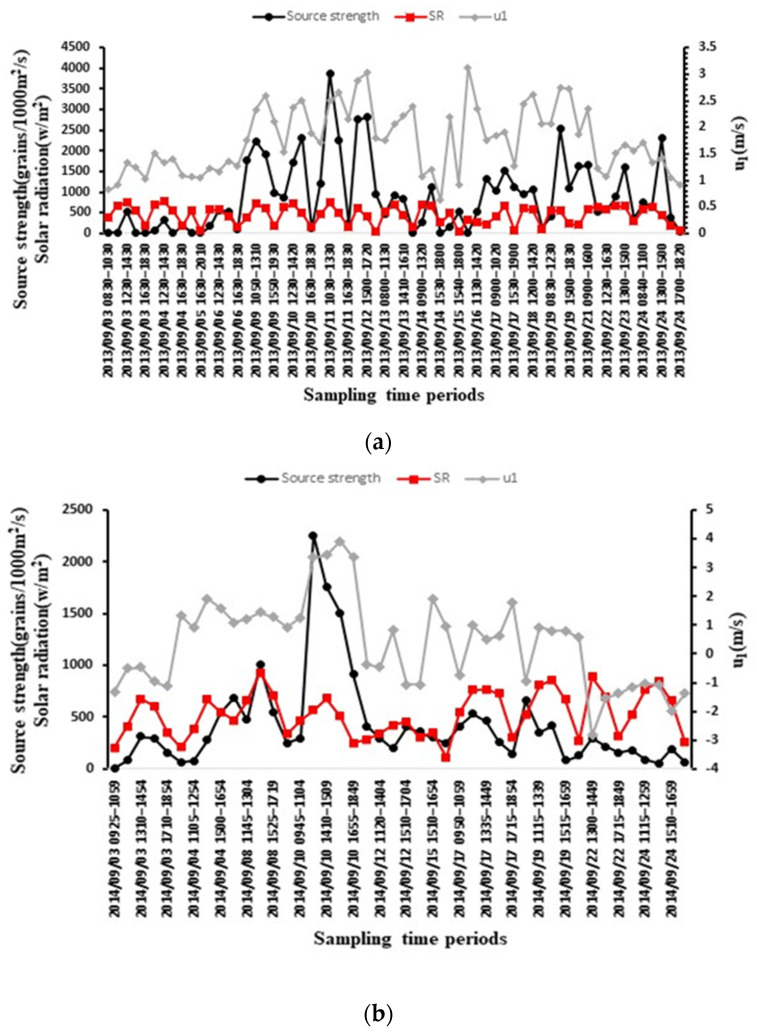
Hourly seed source strength (Q_0_, black line), solar radiation (SR, red line), and the mean wind speed (u1, grey line) from 3 September to 24 September for (**a**) Illinois source field; (**b**) Tennessee source field.

**Figure 11 plants-11-01102-f011:**
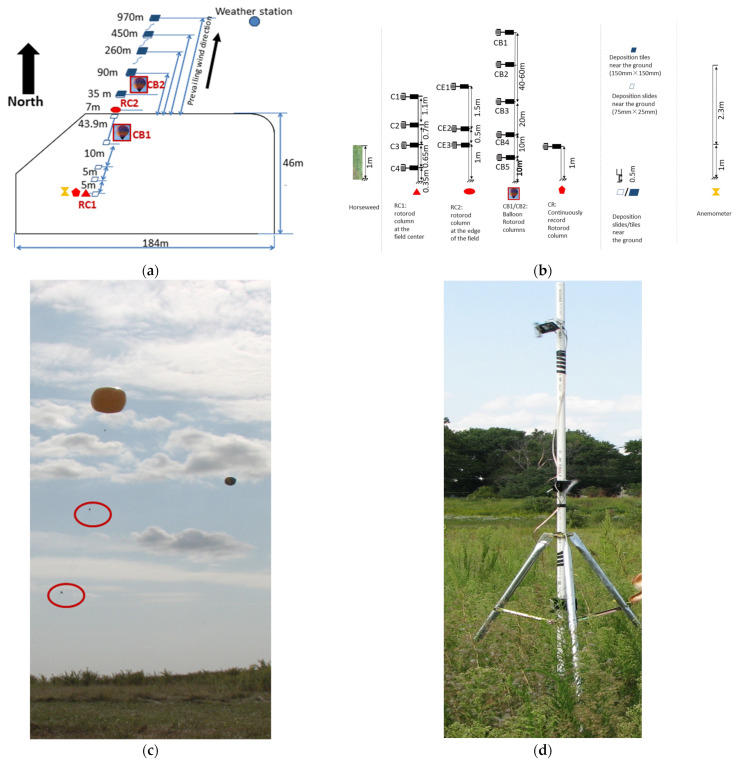
Schematic map and setup of the experiment at the Illinois field: (**a**) Experimental site; (**b**) Sampler heights; (**c**) Floating balloons with Rotorod samplers; (**d**) Rotorod columns. The concentration (grains/$m^{3}$) was measured by two Rotorod columns (RCi, i = 1 to 2) and balloons (CBi, i = 1 to 2). RC1 was placed at the center of the field with four Rotorod samplers (‘Ci’, i = 1 to 4) positioned at different heights. RC2 was set up at the field edge with three Rotorod samplers (‘CEi’, i = 1 to 3). Two columns of Rotorod samplers (‘CBij’, i = 1 to 2, j = 1 to 5) were mounted below two balloons to measure concentration at greater heights (10 to 100 m). In the picture, the Rotorod samplers were marked with red circles. Deposition (grains/$m^{2}$/s) was the data collected by slides and tiles in and outside of the field at the height of 0.5 m above the ground. One additional sampler, i.e., a continuous record Rotorod (CR) and a sonic anemometer were placed in the field.

**Figure 12 plants-11-01102-f012:**
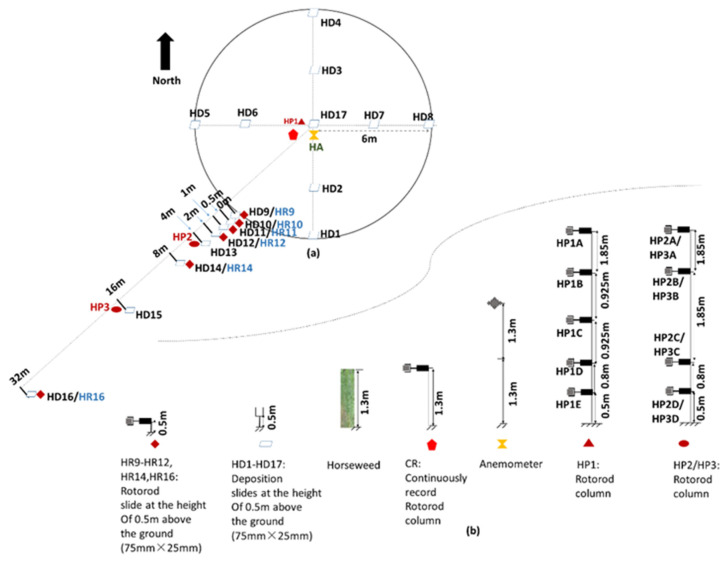
Schematic map and setup of the experiment at the Tennessee field: (**a**) Experimental site; (**b**) Sampler heights. The concentration (grains/$m^{3}$) was measured by three Rotorod columns (HPi, i = 1 to 3) and Rotorod slides at the height of 0.5 m above the ground (HRi, i = 1 to 12,14,16). HP1 was placed at the center of the field, with five Rotorod samplers (‘HP1i’, i = A to E) positioned at different heights. HP2 and HP3 were placed 4 and 16 m away from the edge of the circular field in the downwind direction, with four Rotorod samplers (‘HPji’, j = 2 to 3, i = A to D). Deposition (grains/$m^{2}$ /s) was data collected by slides (75 mm × 25 mm) in and outside of the field at the height of 0.5 m above the ground. In the field, two sampling lines (containing a total of nine sampling locations: HD1 to HD8, HD17) were set up in the direction of east to west and south to north. Outside of the field, there were 8 more deposition slides placed along the downwind direction (HD9 to HD16). An additional sampler (CR) and a sonic anemometer were placed in the field.

**Figure 13 plants-11-01102-f013:**
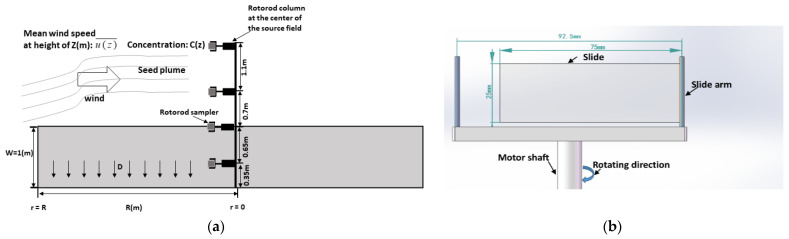
(**a**) Schematic sketch of seed source strength measurement ($Q_{0}$, grains/plant/s) in the source fields, where r was the distance from the Rotorod sampling column at the field center, R is the length of the field edge to the sampling column, in the Tennessee field, R was equal to the radius of the field, C (z) was the concentration (grains/$m^{3}$), $\bar{u\left(z\right)}$ (m/s) was the horizontal mean wind speed, Z (m) was the height of the Rotorod sampler, D represented the downward deposition of seed grains. (**b**) Schematic sketch of the sampling head of the Rotorod sampler. The diameter of the aluminum rotating rod was 92.5 mm. The microscope slides were 25 × 75 mm.

**Table 1 plants-11-01102-t001:** Measurements made and methods used during Illinois and Tennessee experiments.

Measurement	Method	Illinois Year 2013 Z (m)	Tennessee Year 2014 Z (m)
Seed concentration vertical profiles	Slides armed in rotating Rotorod columns	Z = 0.35, 1, 1.7, 2.8 location: the field center	Z = 0.5, 1.3, 2.225, 3.15, 5 location: the field center
		Z = 1, 1.5, 3 location: the field edge	Z = 0.5, 1.3, 3.15, 5 location: 4 m, 16 m from the field edge, 2 columns
		Z = 0–100 location: beneath balloon2 columns	Z = 0.5 Distance = 0, 0.5, 1, 2, 4, 8, 32 m from the field edge location: downwind direction
		Z = 1 location: the field center (continuous record Rotorod)	Z = 1.3 location: the field center (continuous record Rotorod)
Seed deposition	Slides placed on the holders	Z = 0.5 Distance = 0, 5, 10, 20, 63.9 m from the center concentration column location: in the field	Z = 0.5 Distance = 3, 6 m from the center of the circular field location: in the field
		Z = 0.5 Distance = 7, 35, 90, 260, 450, 970 m from the field edge location: downwind direction	Z = 0.5 Distance = 0, 0.5, 1, 2, 4, 8, 16, 32 m from the field edge location: downwind direction

Z, Rotorod sampler height.

**Table 2 plants-11-01102-t002:** Statistics of meteorological variables collected in the Illinois experiment.

Parameter	Symbol	Unit	Height (m)	Source	Mean ± Standard Deviation
Wind direction	Θ (3.3)	degree	3.3	sonic anemometer	228 ± 71
Mean wind speed	$\bar{u_{1}}$(3.3)	m/s	3.3	sonic anemometer	1.84 ± 0.69
Friction velocity	u* (3.3)	m/s	3.3	sonic anemometer	0.36 ± 0.12
Stability	ξ (3.3)	unitless	3.3	sonic anemometer	−2.03 ± 3.75
Air temperature	T	°C	3.3	sonic anemometer	25.42 ± 4.88
Relative humidity	RH	%	2.0	weather station	54.21 ± 14.70
Solar radiation	SR	Kw/m^2^	2.0	weather station	0.43 ± 0.21
Rainfall	Rainfall	mm/hour	2.0	weather station	0.21 ± 2.00

## Data Availability

Data available on request due to restrictions eg privacy or ethical.The data presented in this study are available on request from the corresponding author. The data are not publicly available due to privacy.

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
