# Peer review of "Dynamic Seed Emission, Dispersion, and Deposition from Horseweed (Conyza canadensis (L.) Cronquist)"

_plants, 2022, doi:10.3390/plants11091102_

Round 1
Reviewer 1 Report
Review of Dynamic Seed Emission, Dispersion, and Deposition from 2
Horseweed (Conyza canadensis)
The authors should define glyphosate as being a nonselective systemic herbicide.
In the introduction, the scientific name of the plant is given as Conyza Canadensis but should be Conyza canadensis as in the title. The authors should state the family, Asteraceae, and also mention that this plant was previously, and sometimes still, is known as Erigeron canadensis.
More on the structure and ecology of the plant is needed. The authors should provide a short description of the species, accompanied by photos of the entire plant in bloom, the flower and the seed.
The authors should also tell how the flowers are pollinated and if there are significant insect herbivores.
In Fig.1, show clearer indications of the placement of the two Rotorod columns (RC1, RC2) and balloons placed at the center of the field with the four Rotorod samplers. The present drawings could be supplemented with photographs.
The authors need to add additional information, such as differences of seed release under sunny versus cloudy conditions and in daylight vs night time, etc.
Reviewer 2 Report
Comments and recommendation to MS:
Dynamic Seed Emission, Dispersion, and Deposition from Horseweed (Conyza canadensis)
General comments:
The article presents a study of production and dispersion of seeds of Conyza canadensis which was reported as invasive species in many countries. In general, article is well done and it is worth to be published in Plants after revision.
Specific comments:
Thera are many minor mistakes in the text and figures. So, check carefully the whole text and figures again.
References should be cited according to journal style through all text.
Title
Please, put 'Erigeron canadensis L.' as an accepted scientific name instead of 'Conyza canadensis'
Abstract
Line 14: please, put 'Erigeron canadensis L.' as an accepted scientific name instead of 'Conyza canadensis'. Also, add ‘(L.) Cronquist' after canadensis. It should be: 'Conyza canadensis (L.) Cronquist'. Add note that 'Conyza canadensis (L.) Cronquist' is a synonym of 'Erigeron canadensis L. It is more appropriate to be: 'Erigeron canadensis L. (syn. Conyza canadensis (L.) Cronquist)'
Lines 22, 23: maybe should add space before and after a sign of equality. See journal style.
1. Introduction
Line 41: change into: 'Erigeron canadensis L. (syn. Conyza canadensis (L.) Cronquist)'
Line 42: after ‘worldwide' add on some way that this species is invasive. For example, add 'It is invasive species in some countries'.
- Materials and Methods
Figure 1
Please, try to shorten the text below the picture. This is not a common way of describing an image. For example, remove the sentence: ‘The field had natural horseweed plants at an average height of 1 m.' (this data is also mentioned in the text.).
Also, move some sentences in the textual part, for example: 'One balloon was outside of the source field along the downwind direction, and another balloon was either inside the field or outside along the downwind direction. Balloon horizontal locations and sampler heights on the balloons were adjusted during the experimental period based on if any seeds were detectable at the corresponding sampling heights and locations.'
Figure 2
See comment for Figure 1 (try to shorten the text).
Line 128: delete one full stop after off.
Table 1
It is unusual to put unit for length (m) after each number. So, change that.
Add space after number and sign of equality.
Row 6: 'Z=0-–100m' two hyphen?
Line 146: add space after 'density.'
Line 158: add space after 'detectable.'
Line 168: add space in 'Δ?(?3)'
Line 169: add space in '?(revolution/s)'
Line 170: add space in 'Δ?(s)'
Lines 168, 170, and 171: check font size
Line 210: add space before and after sign of equality in '?Ì… =0' and '?Ì…=0'
Table 2
Row 5: change hyphen '-' into symbol '–'
Figure 3
See comment for Figure 1 (try to shorten the text).
Line 240: add space in 'L(m)'
Line 243: add space in ‘?(?)Ì…Ì…Ì…Ì…Ì…Ì…(m/s)'
There is no information about statistical software which was used for calculations.
Figure 7
Add unit at x-axis
Figure 8
Add space at y-axis (before brackets)
Figure 9
Add unit at y-axis (after concentraion)
Put Figure 9 in Bold.
Figure 10
Add unit at y-axis (after concentraion)
Put Figure 10 in Bold.
Add space after a sign of equality.
Figure 11
Put Figure 11 in Bold.
Figure 11a
Change 'normalized' into 'Normalized'
Line 461: Add space after a sign of equality.
Line 470 and equation no. 11: SS or ss
Figure 12
Put Figure 12 in Bold.
References
Ref. 2,5,7…: Conyza canadensis should be always write on the same way and put in Italic, that is Conyza canadensis
DOI numbers are missing.
